# Potential Control of Mycotoxigenic Fungi and Ochratoxin A in Stored Coffee Using Gaseous Ozone Treatment

**DOI:** 10.3390/microorganisms8101462

**Published:** 2020-09-23

**Authors:** Asya Akbar, Angel Medina, Naresh Magan

**Affiliations:** Applied Mycology Group, School of Water, Energy and Environment, Cranfield University, Cranfield, Bedfordshire MK43 0AL, UK; asyaakbar@live.com (A.A.); a.medinavaya@cranfield.ac.uk (A.M.)

**Keywords:** control, *Aspergillus* section *Circumdati*, section *Nigri*, intervention, ochratoxin A, coffee beans, ozone treatment, storage

## Abstract

The objective of this study was to examine the effect of treatment of Arabica green coffee beans with gaseous ozone (O_3_) for the control of ochratoxigenic fungi and ochratoxin A (OTA) contamination by *Aspergillus westerdijkiae*, *A. ochraceus*, and *A. carbonarius* during storage. Studies included (i) relative control of the populations of each of these three species when inoculated on irradiated green coffee beans of different initial water availabilities using 400 and 600 ppm gaseous O_3_ treatment for 60 min at a flow rate of 6 L^−1^ and on OTA contamination after 12 days storage at 30 °C and (ii) effect of 600 ppm O_3_ treatment on natural populations of green stored coffee beans at 0.75, 0.90, and 0.95 water activity (a_w_) or with additional inoculum of a mixture of these three ochratoxigenic fungi after treatment and storage for 12 days at 30 °C on fungal populations and OTA contamination. Exposure to 400 and 600 ppm O_3_ of coffee beans inoculated with the toxigenic species showed that there was less effect on fungal populations at the lowered a_w_ (0.75). However, toxigenic fungal populations significantly increased 48 h after exposure and when stored at 0.90 and 0.95 a_w_ for 12 days. All three species produced high amounts of OTA in both O_3_ treatments of the wetter coffee beans at 0.90 and 0.95 a_w_. Gaseous O_3_ (600 ppm) treatment of naturally contaminated green coffee beans had little effect on fungal populations after treatment, regardless of the initial a_w_ level. However, after storage, there was some reduction (26%) observed in coffee at 0.95 a_w_. In addition, no fungal populations or OTA contamination occurred in the 0.75 and 0.90 a_w_ treatments after exposure to 600 ppm gaseous O_3_ and storage for 12 days. It appears that under wetter conditions (≥0.90–95 a_w_) it is unlikely that fungal populations and OTA contamination of stored coffee beans, even with such high O_3_ concentrations would be controlled. The results are discussed in the context of potential application of O_3_ as an intervention system for stored coffee post-fermentation and during medium term storage and transport.

## 1. Introduction

Ozone (O_3_) is a highly unstable triatomic molecule which has been shown to have effective germicidal properties. Thus, it has been used in the agri-food industry as an anti-microbial agent to minimise contaminants in raw commodities as it can be lethal to many bacteria and has been used in food packaging applications [1,2]. While natural levels of O_3_ are between 0.01–0.15 ppm, exposure to higher levels can produce some harmful effects on health, especially respiratory and cardiovascular diseases [3,4]. Thus, there are occupational health and safety recommendations including legislative limits for human exposure to O_3_ of 0.1 ppm for 8 h continuously or 0.3 ppm for 15 min [5]. O_3_ is also a very reactive gas and will degrade rubber tubing and other materials readily. Thus, for treatment of raw or processed food matrices the materials used for generating and measuring O_3_ must not be easily corroded (e.g., polytetrafluoroethylene (PTFE) tubing, glass or stainless steel cylinders for treatment of commodities) or steel storage bins or silos.

Fungal species may have a variable sensitivity/tolerance to gaseous O_3_ exposure depending on time of exposure × concentration used [6,7,8]. Sensitivity/tolerance is also influenced by moisture content (m.c.), substrate, and spore morphology [6]. Inactivation of spores and inhibition of mycelial growth, when exposed to gaseous O_3_, have shown variable results. For example, Vijayanandraj et al. [9] reported that *Aspergillus niger* mycelial growth was reduced but that spore germination was unaffected by O_3_ treatment. Interestingly, mycelial growth and sporulation were inhibited by continuous low concentration O_3_ exposure of peaches when inoculated with *Monilinia fructicola*, *Botrytis cinerea*, *Mucor piriformis*, or *Penicillium expansum* and kept in store [10].

There has been a debate as to whether gaseous O_3_ treatment is fungistatic or fungicidal [11,12]. Minas et al. [12] and Hibben and Stotzky [6] reported that germination rates for thick-walled multicellular spores of *Alternaria* and *Stemphylium* spp. were the same when exposed to air or 1 ppm O_3_. However, germination of thin-walled spores of *Rhizopus stolonifer*, *Trichoderma viride*, *A. niger* at the same O_3_ concentration was reduced. O_3_ has been shown to act against unsaturated lipids in the microbial cell membranes causing a leakage of their contents, and subsequent microbial lysis [13,14].

These studies did not examine whether repair mechanisms may be operative and whether there could be a recovery of germinative capacity subsequently. Mylona et al. [15] showed that exposure of *Fusarium verticillioides* conidia to 200–300 ppm O_3_ exposure for 1 h was initially effective but that over the subsequent 10 day period, spore viability recovered and indeed resulted in mycotoxin (fumonisins) production under different a_w_ conditions. Although the inhibition of spore viability by gaseous O_3_ has been examined, the ability for physiological repair probably needs more attention.

Studies with cereals and other food commodities including nuts have suggested that exposure to O_3_ gas can indeed influence contaminant mycotoxigenic mould spores and in some cases reduce mycotoxin contamination, with some claims of effective detoxification [14,16]. For example, Sultan [17] reported that the populations of *Aspergillus flavus* in stored peanuts were significantly decreased after exposure for 1 h at up to 300 ppm. El-Desouky et al. [18] showed that aflatoxin B_1_ (AFB_1_) in wheat was effectively degraded by approximately 55 to 77% after exposure to 40 ppm O_3_ for 5 to 20 min. Sahab et al. [19] found that 40 ppm O_3_ for 10 min led to 94.6 and 99.5% reduction in AFB_1_ and aflatoxin B_2_ (AFB_2_) respectively. Chen et al. [20] suggested that the optimal conditions for detoxification of aflatoxins (AFs) in stored peanuts was by using 6 mg L^−1^ (= 2.8 ppm) for 30 min, with degradation rates of the total AFs and AFB_1_ being approximately 66%. However, many of these studies were carried out in vitro or did not examine differentially how water availabily will influence the efficacy of the O_3_. Mylona et al. [15] certainly found that maize inoculated with *F. verticillioides* and exposed to up to 300 ppm O_3_ for 60 min (4 L min^−1^) and then stored for 15 and 30 days did provide some control of fumonisins contamination at different storage a_w_ levels. However, for some commodities, especially those with a high level of fatty acids/lipids, treatment with high concentrations of O_3_ can have a negative effect on quality of the food by causing oxidation of lipids, changing colour, modifying some vitamins and phenolic compounds and sometimes causing off-odours [14,21]. There have been practically no studies to examine the potential use of gaseous O_3_ for the control of colonisation of coffee by ochratoxigenic species and on potential reduction in ochratoxin A (OTA) contamination during storage of coffee beans under different water availability conditions [22,23]. In contrast, there have been significant studies to examine the breakdown of other mycotoxins, especially AFs and trichothecenes by treatment with gaseous O_3_ in a range of durable commodities, especially temperate cereals [14].

Thus, the aim of this study was to examine the efficacy of different gaseous O_3_ concentrations to reduce or inhibit populations of ochratoxigenic fungi and OTA contamination of stored green coffee by the dominant species found in coffee beans (*A. westerdijkiae*, *A. ochraceus*, and *A. carbonarius*) under different a_w_ levels (0.75, 0.90, and 0.95). Studies were carried out using two systems: (a) irradiated green coffee which was inoculated with spores of each individual ochratoxigenic species and exposed to 400 and 600 ppm O_3_ and then stored for 12 days at 30 °C, and (b) naturally contaminated coffee beans modified to the same three a_w_ levels or that inoculated with a mixture of conidia of these three ochratoxigenic species and exposed to 600 ppm O_3_ and then stored for 12 days at 30 °C. The effects of treatments on fungal populations 48 h after treatment and after 12 days were determined. The OTA contamination was determined after the 12 day storage period.

## 2. Materials and Methods

### 2.1. Fungal Species and Strains Used in These Studies

Strains of two species from the *Aspergillus* section *Circumdati* group, *Aspergillus westerdijkiae* (CBS 121986) and *A. ochraceus* (ITAL 14), and one from the Section *Nigri* group, *A. carbonarius* (ITAL 204) were used in these studies. These were all isolated from green coffee beans and are known as OTA producers [24,25]. The CBS 121986 strain was kindly provided to us by Dr B. Patino (Complutense University, Madrid, Spain) and the others by Dr. M.H. Taniwaki (ITAl, Campinas, Brazil).

### 2.2. Spore Suspensions of the Ochratoxigenic Species for Inoculation of Coffee Beans

The three species were all sub-cultured on 6% green coffee extract and incubated at 25 °C for 7–10 days. 10 mL of sterile water containing 0.01% tween 80 was decanted onto the surface of the cultures. These were agitated with a surface-sterilised bent glass rod to release the conidia. The suspensions were decanted into a 25 mL sterile Universal glass bottle and shaken well. The concentration was quantified with a haemocytometer and diluted with sterile water as required to obtain an inoculum conidial concentration of 10^4^ spores mL^−1^. This was used for the inoculation of the naturally contaminated coffee beans and for the experiments with indiviudal species and irradiated coffee beans.

### 2.3. Development of the Moisture Adsoprtion Curves for Natural and Irradiated Green Coffee Beans

Moisture adsorption curves were constructed to determine the amounts of water necessary for addition to naturally contaminated Arabica green coffee beans, or that which had been gamma irradiated (12–15 kGys; Synergy Healt, Swindon, Berkshire, U.K.) [25]. Known amounts of water were added to 5 g green coffee bean sub-samples and equilibrated at 4 °C for 24 h, then returned to ambient conditions and the water activity (a_w_) of the hydrated green coffee beans was measured. This was done using an Aqualab 4TE (Decagon Devices Inc., Pullman, WA, USA). The coffee bean samples were then dried at 110 °C for 24 h and kept in a desiccator at room temperature for 1 h and weighed to determine the moisture content. The moisture adsorption curves differed between the two types of coffee beans. Thus to obtain the target a_w_ values for natural and irradiated coffee beans for the experiments the amounts of water necessary for 0.75, 0.90, and 0.95 a_w_ treatments were 0.25, 0.9, 2.1 mL, and 0.7, 1.5, and 2.5 mL per 5 g^−1^, respectively.

### 2.4. Ozone Generation, Measurement, and Exposure of Coffee Treatments

All O_3_ treatments were carried out in a fume cupboard with an extractor to avoid user exposure when carrying out these experiments. The system used consisted of a corona discharge ozone generator (C-Lasky series, Model CL010DS, AirTree Ozone Technology Co., Ltd., Sijhih City, Taipei County 22150, Taiwan). The O_3_ generator was was placed outside the fume cupboard and connected via PTFE tubing to the bottom of a glass column via an inlet valve. Up to 50 g of coffee could be placed in the glass column. The glass column was held uprght in a steel laboratory clamp stand in the fume hood. The top of the glass column had an exit bi-valve which was also connected to an O_3_ analyser (Model UV-100, Eco Sensors Inc., Santa Fe, NM 87505, USA). This was to ensure that the exposure of the coffee bean samples was kept at the target O_3_ treatment exposure level during the treatment exposure period (400 or 600 ppm O_3_, 6 L min^−1^; 60 min; 25 °C) accurately. The rest of the O_3_ was vented via the fume hood to the exterior. Only Teflon tubing was used to avoid degradation of the components during experiments.

### 2.5. Effect of Gaseous O_3_ Treatment on Fungal Populations and Ochratoxin A Contamination of Stored Coffee Beans Inoculated with Each Individual Ochratoxigenic Species

The irradiated Arabica green coffee beans (3 × 150 g for each fungal species) were weighed and placed into sterile 500 mL Pyrex glass bottles (Thermo Fisher Scientific Ltd., Basingstoke, Hampshire, UK). The coffee beans were modified to the target a_w_ levels of 0.75, 0.90, and 0.95, excluding 1 mL for the inoculation with the test fungal species. After equilibration overnight, 1 mL of the individual conidial suspension (10^4^ condia ml^−1^) was added separately to each set of treatments. The glass bottles were shaken to mix the conidial inoculum with the coffee beans and left for 1 h. Approximately 45 g was introduced into the glass column in the sterile flow bench and then connected to the O_3_ treatment apparatus which was already set at the target O_3_ concentration generation level. A total of three replicates of each fungal species treatment were exposed to 400 and 600 ppm O_3_ for 60 min at a flow rate of 6 L min^−1^. The control treatments were inoculated and exposed to air only.

After treatment the coffee beans were placed directly into 50 mL surface-sterilised glass containers (Magenta, Sigma-Aldrich Ltd., St Louis, MI, USA) and covered with microporous lids. These were all placed into larger plastic environmental chambers which also contained 2 × 500 mL of a glycerol/water solution in 750 mL glass beakers to maintain the equilibrium relative humidity (ERH) of the atmosphere in the environmental chamber and stored at 30 °C for 12 days.

After 48 h and 12 days, sub-samples were taken and used for quantifying the populations of the inoculated species in relation to the O_3_ treatments. The fungal populations were quantified by placing 10 g in 90 mL of sterile water and shaking well. After serial dilutions, 200 μL of each dilution were spread plated onto three replicate malt extract agar (Thermo Fisher Scientific Oxoid Ltd., Basingstoke, Hampshire, U.K.) plates at different dilutions and incubated at 30 °C for 4–5 days before enumeration. After 12 days, the remaining treatment and replicate samples were all dried and stored at −20 °C until OTA analyses.

### 2.6. Effect of O_3_ on the Total Fungal Populations in Naturally Contaminated Coffee Beans and That Inoculated with a Mixture of the Three Ochratoxigenic Species

#### 2.6.1. Effects of O_3_ on Fungal Populations in Naturally Contaminated Green Coffee Beans

Naturally contaminated coffee beans were rewetted according to the water absorption curve as described previously to obtain the target a_w_ levels of 0.75, 0.90, and 0.95. Three replicates of each treatment (45 g) were exposed for 60 min to 600 ppm O_3_ in the glass column system at a flow rate of 6 L min^−1^. Directly after the treatment with O_3_, approximately 3 g sub-samples were collected from the top, middle, and bottom of the glass treatment chamber as detailed previously, combined, and used for the determination of the total fungal populations.

The rest of the coffee beans were stored for 12 days in the environmental chambers in separate glass containers with microporus lids and containing glycerol/water solutions (500 mls × 2 in glass beakers) to maintain the ERH of the atmosphere in the closed containers. Again, after 12 days storage the fungal populations were determined using the serial dilution method as described previously.

The rest of the sample was dried and stored at −20 °C for later OTA quantification.

#### 2.6.2. Effect of O_3_ on Naturally Contaminated Coffee Beans with Additional Mixed Inoculum of the Three Ochratoxigenic Species

The same a_w_ treatments were prepared (0.75, 0.90, 0.95 a_w_) taking into account the addition of the fungal inoculum. A 5 mL spore suspensions of 10^4^ conidia ml^−1^ was prepared as described previously of each species (*A. westerdijkiae*, *A. ochraceus*, and *A. carbonarius*). They were mixed together to obtain a total of 15 mL. The three replicates of each rewetted naturally contaminated coffee bean treatments were inoculated with 1.5 mL of the mixed spore inoculum and mixed well for 2 min. The replicate coffee bean treatments (45 g) were then placed in the glass column system and exposed to 600 ppm O_3_ at a flow rate of 6 L min^−1^ for 60 min.

The fungal populations were determined after treatment and after storage for 12 days as described previously using serial dilution. The rest of the samples were dried and stored at −20 °C for later OTA quantification.

### 2.7. Ochratoxin A Extraction and Quantification

The dried coffee samples were milled using a Waring Laboratory homogeniser (model 7009G; (Waring Laboratory Science, Torrington, CT, USA)) for 5 min at maximum speed and 10 g of the milled dried coffee beans extracted with a 50 mL methanol:water (70:30) solution in 1% sodium bicarbonate. The extracts were then filtered and 5 mL diluted with 45 mL phosphate buffered saline (PBS/Tween (0.01% *v*/*v*) and applied to an immunoaffinity column (Neogen Europe Ltd., Auchincruive, Ayr, Scotland, UK). 1.5 mL was dried and 0.5 mL of acetonitrile:water (50:50) added. The final extracts were analysed by HPLC (Agilent, Berkshire, UK). The retention time of OTA under the conditions described was approximately 2.5 min. The mobile phases used were acetonitrile (57%): acetic acid (2%): water (41%) [15]. A 20 μL aliquot of the extracted toxin from the treatments and replicates were injected into the HPLC system. The conditions for OTA detection and quantification were as follows:Mobile Phase Acetonitrile (57%): Water (41%): Acetic acid (2%)Column 120CC-C18 column (Poroshell 120, length 100 mm, diameter 4.6 mm, particle size 2.7 micron; 600 Bar)Temperature of column: 25 °CExcitation: 330 nmEmission: 460 nmFlow rate: 1 mL min^−l^Volume of sample injected: 20 µLRetention time: Approximately 2.49 minRun time: 17 minLimit of detection: 0.01 ng g^−1^Limit of Quantification: 0.039 ng g^−1^

### 2.8. Statistical Analyses

Normality of the fungal populations and OTA data for the experimental treatments of a_w_ × O_3_ concentration (400, 600 ppm, or 600 ppm only) × storage time (48 h, 12 days) was checked using the Kolmogorov-Smirnov test. Analysis of data, the factors and response and their interaction were examined by the Kruskal–Wallis (non-parametric) if the data was not normally distributed. For normally distributed data, the datasets were analyzed using a Minitab 16 package (Minitab Inc., 2010. State College, Centre County, PA, USA). All experiments were carried out in triplcate and repeated once. The statistical significant level was set at *p* ≤ 0.05 for all single and interacting treatment factors.

## 3. Results

### 3.1. Effect of O_3_ × a_w_ on Populations and Ochratoxin A Contamination of Irradiated Stored Coffee Beans Inoculated with Individual Inocula of A. westerdijkiae, A. ochraceus, and A. carbonarius

Figure 1 shows the effect of gaseous O_3_ treatment with 400 and 600 ppm (60 min, 6 L min^−1^) on populations of all three species after 48 h and 12 days storage. For *A. westerdijkiae* and *A. ochraceus* at 0.75 a_w_ and exposure to 600 ppm O_3_, there was some decrease in populations after treatment, and also after 12 days storage. In contrast, there was an increase in fungal populations of these two species at both 0.90 and 0.95 a_w_ after exposure to 400 and 600 ppm O_3_ and stored for 12 days when compared to the controls. For the *A. carbonarius* strain there was little difference between the controls and O_3_ treatments shortly after exposure. However, after 12 days storage, the populations in the driest treatment (0.75 a_w_) remained low, while in the 0.90 and 0.95 a_w_ treatments, regardless of O_3_ exposure level, there was a significant increase in the populations of all species isolated from the stored coffee beans.

Statistical analyses using the Kruskal–Wallis test on the effect of the a_w_ × O_3_ concentrations and time on populations of each species isolated from the coffee beans showed that there was a significant effect of O_3_, a_w_, and O_3_ × a_w_ factors. However, there was only a significant reduction effect of O_3_ treatment at 0.75 a_w_, with a significant increase at 0.90 and 0.95 a_w_ at both O_3_ concentrations after 12 days storage.

Figure 2 compares the effect of exposure to 400 and 600 ppm O_3_ on OTA contamination of the stored coffee beans inoculated with the individual ochratoxigenic species at 0.75, 0.90, and 0.95 a_w_ compared to the untreated control. For *A. westerdijkiae* there was a reduction of OTA contamination of about 70 and 32% with 400 and 600 ppm, respectively. In contrast, in the wettest treatment (0.95 a_w_), OTA production was higher in the 400 ppm than the 600 ppm O_3_ treatments. For *A. ochraceus*, OTA contamination was only slightly reduced when compared with the controls at 0.90 a_w_, with about 10% at 400 ppm O_3_. There was no effect of O_3_ treatment at 0.95 a_w_ regardless of O_3_ concentration when compared to the controls. For *A. carbonarius*, OTA contamination was reduced by about ~55 and 80%, respectively.

Table 1 and Table 2 show the statistical analyses of the effects of O_3_ and a_w_ and their interaction on OTA production by the three ochratoxigenic species on coffee beans. This showed that both water stress and O_3_ (600 ppm) significantly affected OTA production by *A. westerdijkiae* and *A. carbonarius* at all a_w_ levels tested. For *A. ochraceus*, OTA production was only controlled at 0.75 a_w_.

### 3.2. Efficacy of O_3_ Treatment on Fungal Populations Isolated from Naturally Contaminated and That Inoculated with a Mixture of the Three Ochratoxigenic Species in Stored Coffee Beans

Figure 3 shows the effect of O_3_ exposure (0 or 600 ppm; flow rate of 6 L min^−1^; 60 min) on the total fungal populations (log_10_ + 1 CFUs g^−1^ dry weight) isolated from naturally contaminated coffee beans and those inoculated with a mixture of the three ochratoxigenic strains modified to 0.75, 0.90, and 0.95 a_w_. The populations were measured 48 h after treatment, and after storage for 12 days at 30 °C.

Exposure of naturally contaminated coffee to O_3_ had little effect on the fungal populations at 0.90 and 0.95 a_w_ isolated when compared to the control. After storage, there was a reduction of 24% at 0.95 a_w_ in the 600 ppm O_3_ treatment when compared to the controls. In addition, no fungal populations were isolated from the 0.75 and 0.90 a_w_ treatments when compared to the untreated controls.

Figure 3 also shows that when the naturally contaminated coffee beans were inoculated with a mixture of the three ochratoxigenic species there was only a slight reduction in the populations isolated after treatment with gaseous O_3_. In contrast, when compared to the controls, no fungi were isolated in 0.75 and 0.90 a_w_ after 12 days storage at 30 °C.

Table 3 shows the statistical effect of treatments using the Kruskal–Wallis test. Exposure to 600 ppm O_3_ had a significant effect, and reduced the fungal populations in naturally contaminated stored coffee beans after treatment at both 0.90 and 0.95 a_w_. In the coffee beans inoculated with the mixture of ochratoxigenic species there was a statistically significant effect of the O_3_ treatment at all three a_w_ levels.

### 3.3. Efficacy of O_3_ Treatment on Ochratoxin A Contamination of Naturally Contaminated Green Coffee Beans and Inoculated with a Mixture of the Three Ochratoxigenic Species Stored for 12 Days

Figure 4 shows the effect of O_3_ treatment on the amount of OTA produced (ng g^−1^) in naturally contaminated coffee beans or inoculated with the mixture of three ochratoxigenic species at the three a_w_ levels and stored at 30 °C in the controlled ERHs for 12 days. OTA contamination of coffee beans with the natural mycobiota was inhibited at 0.75 and 0.90 a_w_ when compared with the controls. However, there was no effect in the wettest coffee bean treatment at 0.95 a_w_.

Where the coffee beans were inoculated with a mixed inoculum of the three ochratoxigenic species and then immediately exposed to O_3_, there was a significant reduction in OTA contamination after storage for 12 days at 0.75 and 0.90 a_w_ treatments when compared to the untreated controls. In addition, there was an 88% decrease in OTA production at 0.95 a_w_ (21.4 ng g^−1^ to 2.6 ng g^−1^). The highest OTA production was observed in the 0.95 a_w_ treatment (Figure 4b).

The statistical effect of the O_3_ treatments are shown in Table 4. The Kruskal–Wallis test showed the significant effect of 600 ppm O_3_ in the 0.75 and 0.90 a_w_ in the naturally contaminated, and in all the a_w_ treatments inoculated with the mixture of the three ochratoxigenic species.

## 4. Discussion

### 4.1. Effect of O_3_ on the Fungal Populations and OTA Production by A. westerdijkiae, A. ochraceus, and A. carbonarius When Inoculated on Irradiated Stored Coffee Beans

The coffee a_w_ levels used in these studies represented the equivalent of approximately 12–14% (0.75 a_w_); 20–22% (=0.90 a_w_), and 30–32% (=0.95 a_w_). These represent relatively safe, intermediate, and wet conditions for storage of such an hygroscopic commodity. This allowed the impacts of O_3_ on fungal populations under different moisture content levels to be examined. The approach was to evaluate O_3_ treatment on potential control of both fungal poulations and OTA contamination of stored green coffee beans. Two O_3_ concentrations, 400 and 600 ppm, were initially used in order to compare efficacy in controlling fungal populations of the three important ochratoxigenic species found in stored coffee beans. Our results showed that treatment with gaseous O_3_ had little effect on fungal populations in relatively dry coffee beans in terms of fungal colonisation (0.75 a_w_), while in the other two treatments with more available water, 0.90 and 0.95 a_w_, the populations increased significantly, even after exposure to O_3_. All three species produced high amounts of OTA in both gaseous O_3_ treatments (400, 600 ppm), especially at 0.90 and 0.95 a_w_. Under conditions which probably do not allow these fungal species to grow (0.75 a_w_), little if any OTA was produced [26,27]. It has been suggested that >20 mg L^−1^ (≥11 ppm) of O_3_ destroys conidia, other asexual spores, and fungal populations [28]. Maeba et al. [29] reported inactivation of pure AFB_1_ by 1.1 mg L^−1^ (= 0.5 ppm) for 5 min. However. Allen et al. [30] suggested that fungal mycelium was less resistant to gaseous O_3_ than conidia. This is surprising, as both Sultan [17] and Mylona et al. [15] found that spores were significantly more sensitive to O_3_ than mycelial extension which was relatively unaffected, regardless of a_w_ levels. However, spore inhibition of *Fusarium verticillioides* by >100–200 ppm O_3_ occurred immediately after exposure but incubation for longer periods showed a recovery of germinative capacity, subsequent growth, and fumonisin production after >10 days [15,16]. Previously, mixtures of gaseous O_3_ (1.25 ppm) and air treatment of wet grains resulted in a suppression of a range of toxigenic *Fusarium* spp. and *Alternaria alternata* in the surface layers of the grain exposed for 2 h a day. Unfortunately this study did not examine effects on mycotoxin contamination [28].

The present study showed that very high doses of gaseous O_3_ (400–600 ppm), regardless of a_w_ level, did not appear to affect OTA biosynthesis, especially when the coffee beans were colonized by *A. carbonarius*. The general tolerance of these ochratoxigenic *Aspergillus* spp. may be due to their darker pigmentation and relatively thick walled conidia which can provide protection against UV-light, sunlight and perhaps gases such as O_3_. Hibben and Stotzky [6] indicated that small hyaline spores are more sensitive to O_3_, while large and pigmented spores, like conidia of *A. niger*, were more resistant [31]. Spores of *A. fumigatus* have been found to be particularly resistant to O_3_ [32]. In contrast, spores of *Fusarium* species (e.g., *F. verticillioides*, *F. langsethiae*) which are practically hyaline and have very little pigmentation appear to be sensitive to O_3_ exposure in air initially, although some recovery of viability was found [15,33]. A thorough comparison amongst species belonging to the same genus is important. In addition, major differences in physiology between different genera may also influence sensitivity/tolerance to O_3_ treatment. It is also very important to consider the relationship between O_3_ concentration × time of exposure for different commodities to maximize efficacy [14,15,28,34].

### 4.2. Efficacy of O_3_ on Fungal Populations and OTA Contamination of Naturally Contaminated Coffee Beans and Inoculated with Ochratoxigenic Species at Different Water Availabilities

Based on these results, 600 ppm gaseous O_3_ was used to compare the potential for control of fungal populations and OTA contamination by resident ochratoxigenic fungi present in the green coffee bean phyllosphere and when this was increased by addition of a mixture of the three key toxigenic species found in the post-fermentation phase of coffee processing. There have been few if any studies on the effects of O_3_ on controlling fungal populations or OTA contamination during storage and transport through different climatic regions to the markets. After O_3_ exposure, there appeared to be very little or no impact on the fungal populations, regardless of initial coffee a_w_ treatment. However, after storage, there was some reduction (26%) observed in the wettest coffee treatment at 0.95 a_w_ + 600 ppm gaseous O_3_. In addition, no fungal populations and OTA contamination was found at 0.75 and 0.90 a_w_ when the coffee was treated with 600 ppm gaseous O_3_ for 60 min. It appears that under wetter conditions (≥0.95 a_w_) and thus a higher relative humidity, it is more difficult to control fungal populations and OTA contamination with O_3_.

Nascimento et al. [22] examined both O_3_ and ultrasound for the treatment of coffee beans during the post-fermentation phase. They showed that the main potential effect was to reduce the total fungal populations present after the fermentation phase. Unfortunately, this study lacks clarity in terms of the actual O_3_ concentrations used, coffee moisture contents, and was focused more on physiological characteristics and brew quality than on mycotoxin contamination. Some studies suggest that in environmentally stressed conditions, O_3_ treatment may be more effective in controlling fungal contamination and perhaps mycotoxin production. For example, Mylona et al. [14] exposed maize grains in situ to gaseous O_3_ (100, 200 ppm) for 60 min and subsequently stored the grain initially inoculated with *F. verticillioides* for 15 and 30 days at marginal conditions for germination and growth (0.88, 0.92 a_w_). Fungal populations, directly after exposure, were significantly reduced and fumonisins, especially at 0.88 a_w_, were minimal. After storage, very low amounts of fumonisin B_1_ were present in the 0.92 a_w_ treatment. Similarly Mylona [33] studied effects of O_3_ on *F. langsethiae* which produces T-2 and HT-2 toxin in oats and found that the production of these toxins was reduced by 38–99% at different a_w_ levels when treated with 200–400 ppm gaseous O_3_. In addition, 74–100% reduction of OTA contamination was observed in naturally contaminated wheat grain inoculated with *Penicillium verrucosum* after treatment with this O_3_ range and stored for 30 days [33].

In the present study, similar effects were found in naturaly contaminated coffee beans inoculated with a mixture of the three ochratoxigenic species. When compared to the controls after storage there was a significant reduction in fungal populations after treatment with 600 ppm O_3_. Complete inhibition of both fungal populations and OTA occurred at both 0.75 and 0.90 a_w_ when treated with this concentration for 60 min. Some studies have used lower doses of O_3_ (75 mg L^−1^ = 35 ppm) and longer exposure times to decrease fungal growth in grains [35]. In other studies, such as those by Prudente and King [36], exceptionally high concentrations of 10 to 12% (by weight = 10.000–12,500 ppm) of O_3_ gas were used which degraded 92% of aflatoxins in corn exposed at a flow rate of 2 L min^−1^ for 96 h. White et al. [37] suggested that between 1000–15,000 ppm of gaseous O_3_ can reduce fungal growth in high moisture maize at 0.5 L min^−1^ treatment for 1 h. These high concentrations may be unfeasible because they are not very user friendly, in the context of the toxicity of O_3_ and the low legal exposure limits detailed previously. It may be better to consider the three key factors: O_3_ concentration level × time of exposure and commodity water availability and use the right combination for optimizing the control required. Certainly, lower concentrations for longer periods of time may be preferable, especially for commodities where food safety as well as quality and flavour may be impacted. This could affect the economic value of such treatments.

## 5. Conclusions

This study has shown that potential does exist for the use of O_3_ gas as a method of reducing fungal populations and OTA contamination of stored green coffee. By examining both the effect of individual fungi and naturally colonised coffee beans it was possible to identify conditions which may lead to effective reductions in OTA contamination. However, this is dependent on the initial moisture content which probabaly needs to be <20% as has been highlighted in this study. The O_3_ concentration (ppm) × time of exposure (h) and perhaps also the type of coffee being treated needs to be further optimised to ensure that there are little if any effects on quality and aroma, which are important considerations. In addition, the costs of O_3_ generation and having the right practical facilities for carrying out the treatment of coffee on a larger scale as well as user safety are important considerations in the utilisation of this approach for this commodity.

## Figures and Tables

**Figure 1 microorganisms-08-01462-f001:**
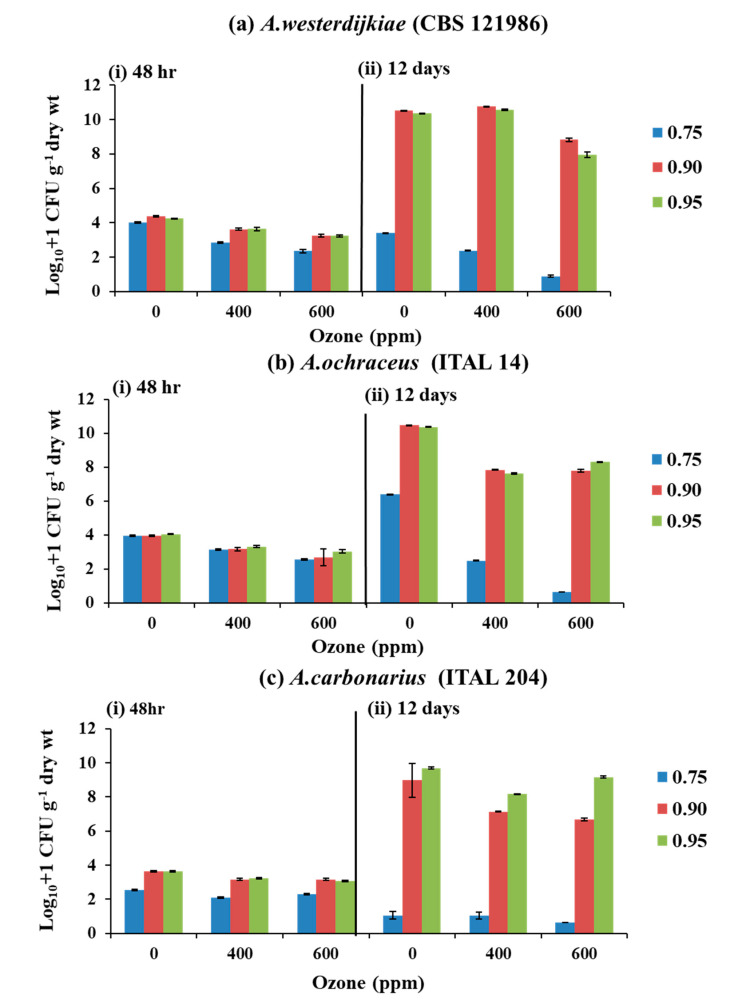
Effect of 0, 400, and 600 ppm O_3_ exposure (60 min; 6 L min^−1^) on the log_10_ + 1 populations of (**a**) *A. westerdijkiae*, (**b**) *A. ochraceus*, and (**c**) *A. carbonarius* pre-inoculated on irradiated coffee beans adjusted to 0.75, 0.90, and 0.95 a_w_, compared to the control (untreated, 0 ppm); (i) 48 h after exposure and (ii) after 12 days storage at 30 °C. Vertical bars indicate the standard error of the means.

**Figure 2 microorganisms-08-01462-f002:**
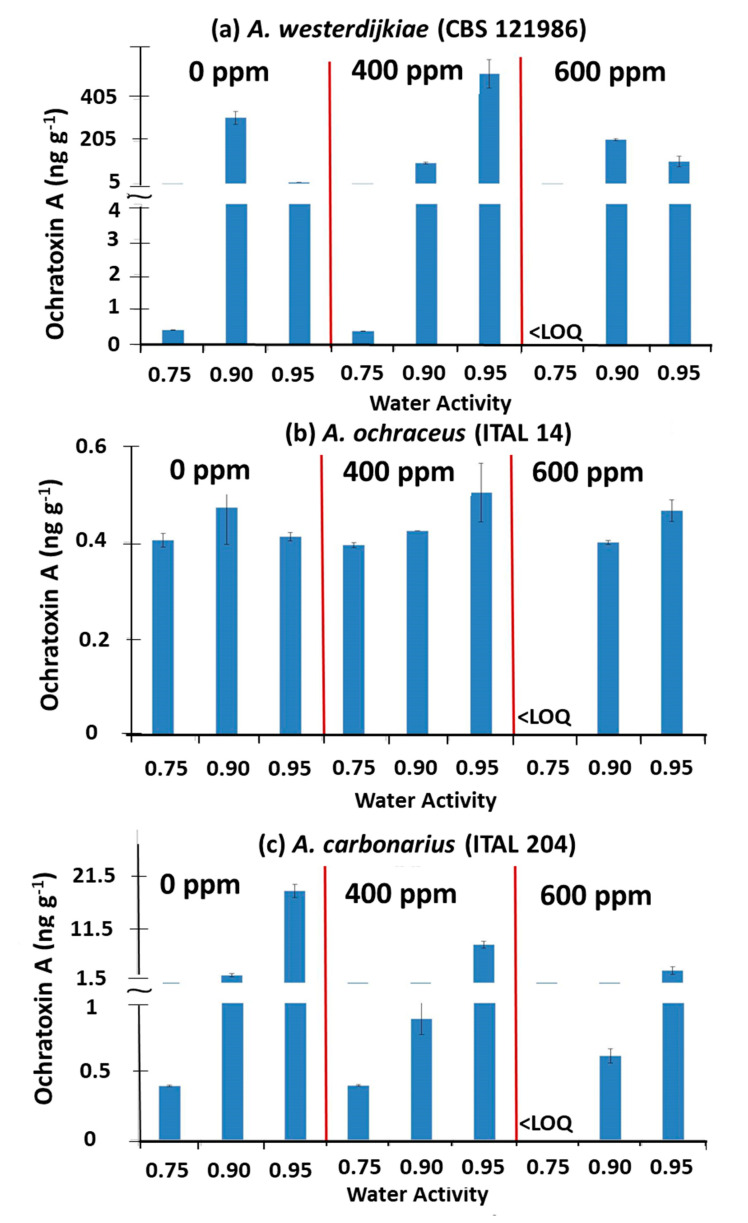
Combined effect of O_3_ dose and a_w_ (0.75, 0.90, 0.95 a_w_) on ochratoxin A production in coffee beans inoculated with (**a**) *A. westerdijkiae*, (**b**) *A. ochraceus*, and (**c**) *A. carbonarius* and treated with O_3_ (400, 600 ppm; 6 L min^−1^, 60 min) and stored at 30 °C for 12 days. Vertical bars indicate the standard error of the means. Key: <LOQ. Less than Limit of quantification.

**Figure 3 microorganisms-08-01462-f003:**
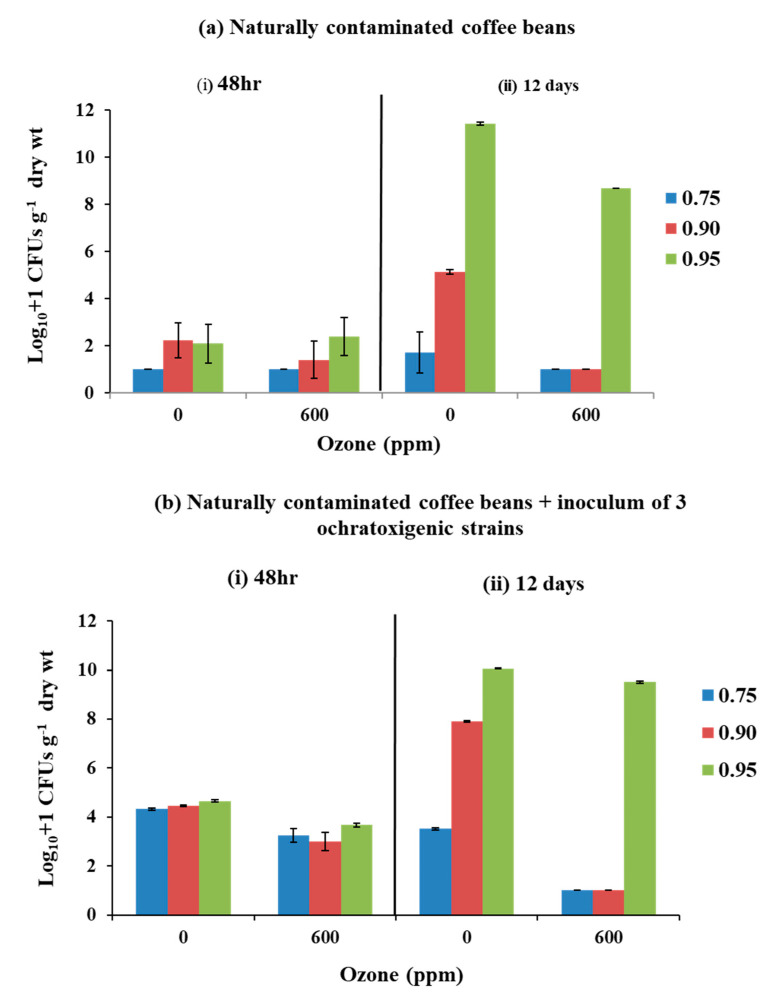
Effect of air or 600 ppm O_3_ exposure for (600 ppm; 6 L min^−1^, 60 min) on the populations (log_10_ + 1 CFUs g^−1^ dry weight) isolated from (**a**) naturally contaminated coffee beans and (**b**) naturally contaminated + inoculum of condia of *A. westerdijkiae*, *A. ochraceus*, and *A. carbonarius* in coffee beans adjusted to 0.75, 0.90, and 0.95 a_w_, compared to the controls (0 ppm O_3_). (i) after exposure and (ii) 12 days storage at 30 °C. Vertical bars indicate the standard error of the means.

**Figure 4 microorganisms-08-01462-f004:**
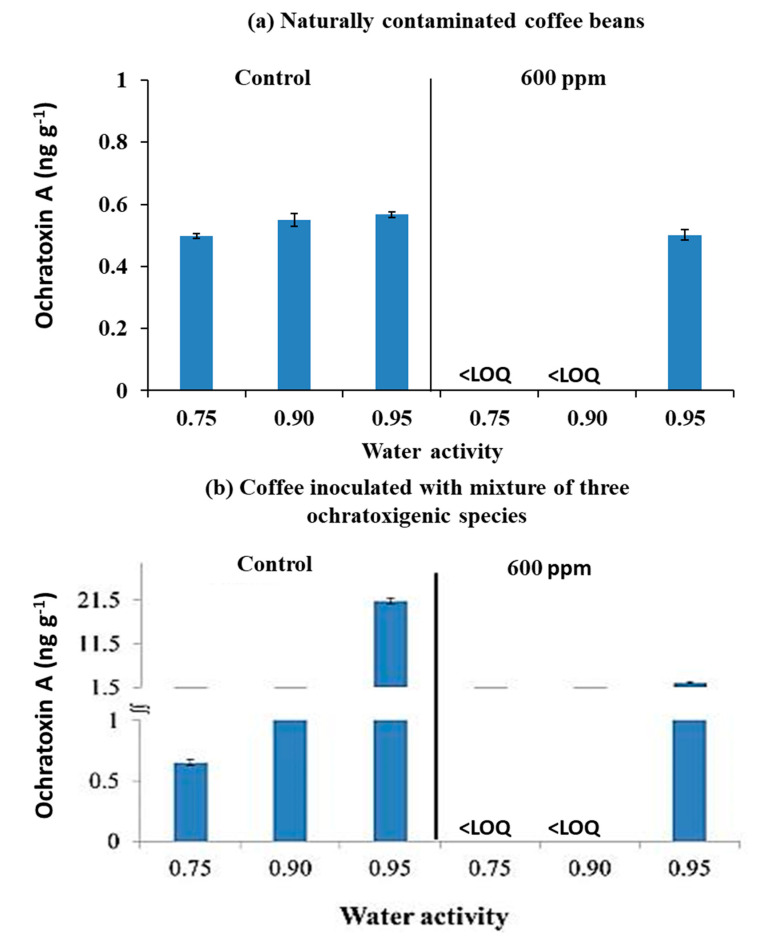
Combined effect of 600 ppm O_3_ × a_w_ (0.98 and 0.94) on ochratoxin A production in (**a**) naturally contaminated coffee beans and (**b**) coffee beans inoculated with conidia of 3 ochratoxigenic species when stored for 30 °C for 12 days. Vertical bars indicate the standard error of the means. Key: <LOQ, less than Limit of Quantification.

**Table 1 microorganisms-08-01462-t001:** Summary of the statistical results of O_3_ treatment on ochratoxin A (ng g^−1^) production by *A. westerdijkiae*, *A. ochraceus*, and *A. carbonarius* in relation to a_w_ (0.75, 0.90, 0.95) after 12 days storage using the Kruskal–Wallis Test (non-normality data).

	Factors	
Strains	a_w_	Ozone (ppm)
*A. westerdijkiae* (CBS 121986)	0.75	S
	0.90	S
	0.95	S
*A. ochraceus* (ITAL 14)	0.75	S
	0.90	NS
	0.95	NS
*A. carbonarius* (ITAL 204)	0.75	S
	0.90	S
	0.95	S

**Table 2 microorganisms-08-01462-t002:** Statistical comparison of ochratoxin A production by the species in relation to O_3_ and a_w_ × O_3_ using ANOVA *.

	Ozone (ppm)	a_w_	a_w_ × Ozone
*A. carbonarius* (ITAL 204)	S	S	-
*A. westerdijkiae* (CBS 121986)	S	S	-
*A. ochraceus* (ITAL 14)	NS	S	-

* *p* ≤ 0.05. at least one signifiant effect in the model; S, Significant; NS, not significant.

**Table 3 microorganisms-08-01462-t003:** The statistical results for control of fungal populations (log_10_ + 1 CFUs g^−1^ dry weight) on (a) naturally contaminated coffee beans and that inoculated with the three ochratoxigenic strains, and (b) compares naturally contaminated coffee beans and those contaminated with the 3 strains by using Kruskal–Wallis Test * (non-normality data).

	Factors					
**(a)**	**a_w_**	**Ozone**	**Days**	**Ozone (0, 600 ppm)**	**a_w_ (0.75,0.90,0.95)**	**Days (48 h, 12 days)**
Naturally contaminated coffee beans (1)	0.75	NS	NS	S	S	S
	0.90	S	NS			
	0.95	S	S			
Coffee beans inoculated with 3 ochratoxigenic strains (2)	0.75	S	S	S	S	S
	0.90	S	S			
	0.95	S	S			
**(b)**	**a_w_**	**Ozone**	**Days**	**Ozone (0, 600 ppm)**	**a_w_ (0.75,0.90,0.95)**	**Days (48 h, 12 days)**
(1) versus (2)	0.75	S	NS	S	S	S
	0.90	S	NS			
	0.95	S	S			

* *p* ≤ 0.05 for at least one significant effect in the model; S Significant; NS Not significant.

**Table 4 microorganisms-08-01462-t004:** The statistical analysis of the effect of O_3_ treatment on ochratoxin A (ng g^−1^) production in (a) naturally contaminated coffee beans and that inoculated with conidia of 3 toxigenic species when compared with the untreated controls and (b) comparison with control of OTA production in naurally contaminated and that inoculated with a mixture of the three species in relation to O_3_ and a_w_ × O_3_ using ANOVA *.

		Factors		
**(a)**	**a_w_**	**Ozone**	**Ozone (0, 600 ppm)**	**a_w_ (0.75,0.90,0.95)**
Naturally contaminated coffee beans (1)	0.75	S	S	NS
	0.90	S		
	0.95	NS		
Coffee beans inoculated with 3 ochratoxigenic strains (2)	0.75	S	S	S
	0.90	S		
	0.95	S		
**(b)**	**a_w_**	**Ozone**	**Ozone (0, 600 ppm)**	**a_w_ (0.75,0.90,0.95)**
(1) versus (2)	0.75	S	S	S
	0.90	S		
	0.95	S		

* *p* ≤ 0.05, at least one significant effect in the model; S Significant; NS Not significant.

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
