# Peer review of "Potential Control of Mycotoxigenic Fungi and Ochratoxin A in Stored Coffee Using Gaseous Ozone Treatment"

_microorganisms, 2020, doi:10.3390/microorganisms8101462_

Round 1

Reviewer 1 Report

General: The work described in the manuscript is original and important to  the safety of stored coffee beans and the health of coffee consumers. Although the fairly manuscript is well written, there are some typo, grammar and writing issues to be addressed before accepting for publication. Specific comments are shown below.   

  1. line 97: change was to were.
  2. line 10: change to " The objective of this study was"
  3. line 108: change "10 mls" to ten milliliters"
  4. line 132-134: It is unclear where the coffee bean were placed in. the glass column or laboratory clamp?
  5. line 138: change "extracted" to "vented"
  6. line 140: change to "fungal populations"
  7. line 144: aw; line 146: 104

Results:

  1. The description of results is wordy and there are some grammar errors.
  2. line 221-222: change "whne inoculated with" to "inoculated with ..., A. carbonarius"
  3. . After read the whole manuscript, I found that the immediately after exposure to O3 was actually 4 hours after exposure. Immediately means right after, that is the same day after treatment. please correct through the manuscript
  4. Line 247-248: Grammar error.
  5. line 273: should it be immediate after exposure?

Discussion

  1. delete the detailed description of results which have been done in the results section.
  2. Discuss the effectiveness of your treatments and whether the treatment conditions (moisture and temperature) used in your study effective enough to control fungal growth/population and ochratoxin A level during storage under the common storage conditions of green coffee beans.
  3. line 326-327: Is the sentence completed?
  4. Line 330-335: This is the repeated description of results.
  5. line 361-364: This long sentence is very confusing.
  6. Line 391-392: change to "when treated at this concentration for 60 min"
  7. Line 399: change "unrealistic" to " impractical or unfeasible"
  8. line 400: delete the repeated three.
  9. line 402-404: good point but the clarity needs to be improved.
  10. line 408: add "of" before stored green coffee.
  11. line 411: what is m.c.? spell it out. "The concentration x time of exposure" describe it using proper sentence.

Author Response

Open Review

Thank you for your comments which will improve the manuscript and its clarity. We have tried to address all the points raised.

With best wishes,

Naresh

Prof. N. Magan

Comments and Suggestions for Authors

General: The work described in the manuscript is original and important to the safety of stored coffee beans and the health of coffee consumers. Although the fairly manuscript is well written, there are some typo, grammar and writing issues to be addressed before accepting for publication. Specific comments are shown below.  

We apologise for the many minor errors and mistakes in the manuscript we have now made every effort to address these.

line 97: change was to were.

Done

line 10: change to " The objective of this study was"

Done

line 108: change "10 mls" to ten milliliters"

Modified based on Reviewer 3 comments

line 132-134: It is unclear where the coffee bean were placed in. the glass column or laboratory clamp?

Modified and reworded for better clarity

line 138: change "extracted" to "vented"

Done

line 140: change to "fungal populations"

Done

line 144: aw; line 146: 104

Done

Results:

The description of results is wordy and there are some grammar errors.

line 221-222: change "whne inoculated with" to "inoculated with ..., A. carbonarius"

We have tried to improve this for better accessibility and clarity.

. After read the whole manuscript, I found that the immediately after exposure to O3 was actually 4 hours after exposure. Immediately means right after, that is the same day after treatment. please correct through the manuscript

This has been changed. Because each treatment with O3 was for were for 1 hr. We had to stagger the experiments to ensure we could complete the replication prior to fungal population quantification and subsequent storage.

Line 247-248: Grammar error.

Modified now.

line 273: should it be immediate after exposure?

Changed to take account of previous comments.

Discussion

delete the detailed description of results which have been done in the results section.

We don’t believe there is a significant amount of detail of the results in the Discussion. We have tried to highlight the interesting ones and discuss this in the context of existing published data. We have slightly modified the Discussion and hopefully this will be adequate.

Discuss the effectiveness of your treatments and whether the treatment conditions (moisture and temperature) used in your study effective enough to control fungal growth/population and ochratoxin A level during storage under the common storage conditions of green coffee beans.

line 326-327: Is the sentence completed?

Completed now

Line 330-335: This is the repeated description of results.

Slightly modified

line 361-364: This long sentence is very confusing.

We have tried to modify to make it better.

Line 391-392: change to "when treated at this concentration for 60 min"

Done

Line 399: change "unrealistic" to " impractical or unfeasible"

Done

line 400: delete the repeated three.

Done

line 402-404: good point but the clarity needs to be improved.

We have tried to improve the clarity

line 408: add "of" before stored green coffee.

line 411: what is m.c.? spell it out. "The concentration x time of exposure" describe it using proper sentence.

We have spelled it out but it is mentioned previously in the methods section in full and then in parentheses (m.c.).

Reviewer 2 Report

The authors studied the impact of ozone treatment on the growth of different fungal strains on green coffee beans adjusted to specific water activities. Additionally, they recorded the formation of ochratoxin A and a possible reduction of ochratoxin A levels based on ozone treatment. The experimental design appears sound and the results are clearly described. The experiments are generally well described. Surprising to me are the very low standard deviations observed in the reported experiments for CFU as well as for OTA formation. The authors should state if the experiments were performed in parallel or, more appropriate for studies with microorganisms, in three individual incubations / passages?

I recommend a more extensive critical discussion with respect to applicability, considering the current techniques and recommendations for coffee storage. See for instance Wintgens (Ed.) Coffee growing, processing, sustainable production: “The generally recognized ideal moisture content of coffee is 12% for Robusta and 13% for Robusta…” so far below the 20% discussed by the authors.

Minor remarks:

L 13: what does “60 mins/6 L−1” mean?

L111: sterile

L118: kGys

L133: and following: g not gms

L146: 104 conidia?

L148 and following: h nor hr

L156: mL not mls

L160: sterile

L169: were

L191: specify “milled”

L196 Agilentmt?

Author Response

Reviewer 2

We thank the Reviewer for the useful comments which will improve the manuscript. We appreciate that there are too many minor errors and have tried to now address these.

With best wishes,

Naresh

Prof. N. Magan

Comments and Suggestions for Authors

The authors studied the impact of ozone treatment on the growth of different fungal strains on green coffee beans adjusted to specific water activities. Additionally, they recorded the formation of ochratoxin A and a possible reduction of ochratoxin A levels based on ozone treatment. The experimental design appears sound and the results are clearly described. The experiments are generally well described. Surprising to me are the very low standard deviations observed in the reported experiments for CFU as well as for OTA formation. The authors should state if the experiments were performed in parallel or, more appropriate for studies with microorganisms, in three individual incubations / passages?

The experiments were carried out with three different glass columns, which were then attached to the O3 fumigation system, one at a time. They were cleaned before isopropanol and dried before being used for another treatment or replicate. The storage for the replicates at specific aw levels were stored in separate chambers and incubated at 30oC.  The log+1 transformation was made of the fungal populations data and a combination of immune-affinity column + HPLC analyses was used for all the OTA quantification.  The SEs varied depending on the treatments and do vary. This is what we obtained.   

I recommend a more extensive critical discussion with respect to applicability, considering the current techniques and recommendations for coffee storage. See for instance Wintgens (Ed.) Coffee growing, processing, sustainable production: “The generally recognized ideal moisture content of coffee is 12% for Robusta and 13% for Robusta…” so far below the 20% discussed by the authors.

We have used a range of m.c. to represent good, intermediate and poor levels of storage. This is often due to uneven sun drying prior to bagging or placing stores. Of course coffee is also a hygroscopic commodity, which when dried to a safe m.c. may reabsorb moisture form the atmosphere. In addition, since OTA contamination is a reality in coffee then often in the chain storage conditions have to have been conducive for the relevant mycotoxigenic species, which usually produce OTA > 15-16% m.c.

Indeed, in the relevant Chapter of the above book it was pointed out that (Robusta) coffee growers in Indonesia and Uganda deliver green (hulled) coffee to the mills. The moisture of green coffee usually exceeds 12% and requires additional drying in the mill itself. It is not unusual for coffee to arrive with a moisture content of 15-20%. This means that the mills need to have large capacity dryers to reach the target safe m.c.s.

Minor remarks:

L 13: what does “60 mins/6 L−1” mean?

We have modified this. This means fumigation for 60 mins with the O3 concentration at a flow rate of 6 l/min. 

L111: sterile

L118: kGys

L133: and following: g not gms

L146: 104 conidia?

L148 and following: h nor hr

L156: mL not mls

L160: sterile

L169: were

All these changes have been made.

L191: specify “milled”

This has been qualified and expanded for clarity.

L196 Agilentmt?

Changed

Reviewer 3 Report

This manuscript an be accepted for publication in Microorganisms upon minor revision.

Specific comments are reported in the annotated version.

The manuscript contains several typos, and I've found it quite annoying to have to mark them. Particularly the youngest authors should bear in mind that correcting typos and language mistakes is not part of the peer reviewer's, nor of the editor's duty: may I suggest that all their future manuscripts shall be carefully checked before submission in order to avoid irritating those who have to consider the suitability for publication.

Author Response

Reviewer 3

We thank the reviewer for the annotated version of the manuscript with the recommended changes. 

Naresh

Prof. N. Magan

The manuscript contains several typos, and I've found it quite annoying to have to mark them. Particularly the youngest authors should bear in mind that correcting typos and language mistakes is not part of the peer reviewer's, nor of the editor's duty: may I suggest that all their future manuscripts shall be carefully checked before submission in order to avoid irritating those who have to consider the suitability for publication.

We have made all the recommended changes in the revised manuscript. We hope these are satisfactory.